# AB_5_-Type Toxin as a Pentameric Scaffold in Recombinant Vaccines against the Japanese Encephalitis Virus

**DOI:** 10.3390/toxins15070425

**Published:** 2023-06-29

**Authors:** Jina Ahn, Ji Eun Yu, Hanna Kim, Jemin Sung, Gyoonhee Han, Myung Hyun Sohn, Baik-Lin Seong

**Affiliations:** 1The Interdisciplinary Graduate Program in Integrative Biotechnology & Translational Medicine, Yonsei University, Incheon 21983, Republic of Korea; 2Department of Biotechnology, College of Life Science and Biotechnology, Yonsei University, Seoul 03708, Republic of Koreahnkim9520@gmail.com (H.K.);; 3Department of Integrated OMICS for Biomedical Science, WCU Program of Graduate School, Yonsei University, Seoul 03722, Republic of Korea; 4Department of Pediatrics, College of Medicine, Yonsei University, Seoul 03722, Republic of Korea; 5Department of Microbiology, College of Medicine, Yonsei University, Seoul 03722, Republic of Korea; 6Vaccine Innovative Technology ALliance (VITAL)-Korea, Yonsei University, Seoul 03721, Republic of Korea

**Keywords:** chaperna, Japanese encephalitis virus, protein folding, recombinant vaccine

## Abstract

Japanese encephalitis virus (JEV) is an enveloped icosahedral capsid virus with a prime neutralizing epitope present in E protein domain III (EDIII). E dimers are rearranged into a five-fold symmetry of icosahedrons. Cholera toxin B (CTB) and heat-labile enterotoxin B (LTB) of AB_5_-type toxin was used as the structural scaffold for emulating the pentameric axis of EDIII. We produced homo-pentameric EDIII through the genetic fusion of LTB or CTB in *E. coli* without recourse to additional refolding steps. Harnessing an RNA-mediated chaperone further enhanced the soluble expression and pentameric assembly of the chimeric antigen. The pentameric assembly was validated by size exclusion chromatography (SEC), non-reduced gel analysis, and a GM1 binding assay. CTB/LTB−EDIII chimeric antigen triggered high neutralizing antibodies against the JEV Nakayama strain after immunization in mice. Altogether, our proof-of-principle study creating a JEV-protective antigen via fusion with an AB_5_-type toxin as both a pentameric scaffold and a built-in adjuvant posits the bacterially produced recombinant chimeric antigen as a cost-effective alternative to conventional inactivated vaccines against JEV.

## 1. Introduction

Vaccines remain the most effective means to prevent and control infectious diseases. Conventional vaccines, such as live-attenuated and inactivated vaccines, rely on cell culture systems, often requiring a high level of safety containment and sophisticated manufacturing facilities, limiting their deployment in endemic areas of low- and middle-income countries (LMICs) [1,2]. Bacterial production of a recombinant protein vaccine would obviate the need for cultures of infectious viruses, thus simplifying the manufacturing process. Advanced technologies, including immunological proteomics, structural vaccinology, and genetic engineering, further contribute to the versatility and customizability when designing a protein vaccine and, hence, improve its efficacy [3,4,5,6]. The most crucial factor in manufacturing a potent recombinant vaccine is to ensure the proper folding of the vaccine antigen into an immunologically relevant conformation. Further, the immunogenicity of recombinant antigen vaccines should be enhanced by combining them with optimal adjuvants or carrier molecules, or by targeting specific immune cells with protein antigens [7,8]. The ability to assemble monomeric proteins into a higher order increases the chances of the antigen being presented to immune cells. Furthermore, increasing the size of an antigen and emulating the viral structure positively impact vaccine efficacy [9].

As one of the prevailing paradigms of virion assembly in both naked and enveloped viruses, the icosahedral capsid is a 20-sided geometric shape composed of equilateral triangles on each side. Icosahedrons can rotate around an axis in a variety of directions, referred to as the 2-3-5 symmetry, composed of two-, three-, and five-fold axes. These viruses increase their capsid size by multiplying structural units in each face of the icosahedron [10]. 

Belonging to the flavivirus family, JEV, in particular, presents a classic example of a virus that has icosahedral symmetry. JEV is a prevalent cause of viral encephalitis in LMICs. Individuals with JEV rarely exhibit symptoms, but those who do risk a high mortality rate of up to 25–30% [11,12]. Western Pacific regions and 24 countries in Asia are still at risk of JEV transmission [13]. Multiple JE vaccines exist but are not universally available in Asia due to licensing, safety, and economic issues [14]. By leveraging bacterial production combined with removal of endotoxin, LMICs can benefit from cost-effective and scalable production, rapid development, ease of downstream processing, and the potential for local manufacturing. Developing a low-cost vaccine, produced preferably by a microbial system, could be a promising approach to reducing the financial burden on vaccine production and delivery. 

There are four types of JE vaccines currently available: (1) inactivated mouse brain-derived vaccine (MBDV); (2) primary hamster kidney-cell-derived, live-attenuated vaccine; (3) inactivated Vero cell-derived vaccine; and (4) live-attenuated chimeric JE vaccine [15,16]. Regardless of type, all four are based on virus culture and come with disadvantages in terms of reactogenicity and affordability. In MBDVs, impurities from the mouse brain pose a safety concern. There are also safety and financial issues associated with attenuated vaccines adopted in mammalian cells [16,17]. Formalin-inactivated vaccines have compromised efficacy due to the chemical modification of protective epitopes in the pentameric configuration of EDIII [18], a particular drawback that emphasizes the preservation of antigenic structure in the pentameric interface as key to protective immunogenicity. These disadvantages of conventional virus cultures necessitate a new innovative platform that ensures low-cost delivery of a safe and effective vaccine.

In JE vaccine development efforts, the enveloped (E) protein is considered the prime target for a protective immune response. The E protein is comprised of 3 domains, EDI, EDII, and EDIII, amongst which, EDIII is involved in virus attachment to the host and stimulating neutralization antibodies [19]. EDIII forms dimers, and they are further packaged along a five-fold symmetry on the viral surface [20], providing a pentameric interface. Thus, emulating the pentameric EDIII of an infectious virion, displaying its native structure and epitopes, would be a critical element in the development of an effective subunit of a vaccine against JEV. 

Here, we present a pentameric assembly platform of viral antigen in an *E. coli* expression system. Our approach was to emulate the five-fold symmetry of EDIII using cholera toxin B (CTB) and heat-labile enterotoxin B (LTB) from the AB_5_-type toxin family [21]. CTB and LTB share a common sequence of 80% [22] and both are known potent mucosal adjuvants [23,24]. As the non-toxic component (B_5_) of a holo-toxin (AB_5_), CTB provides the pentamer scaffold for the assembly of the EDIII domain, emulating the native five-fold axis [25,26]. We hypothesized that the genetic fusion with CTB or LTB would facilitate the pentameric self-assembly of JEV EDIII, retaining the conformational epitopes that generate a proper protective immune response [16,17]. Using the non-toxic component of an AB_5_-type toxin as the structural scaffold and built-in adjuvant, we successfully demonstrated the pentameric assembly of soluble chimeric CTB−JEV EDIII and LTB−JEV EDIII in an *E coli* expression system, and potent neutralizing antibody titers against the JEV infectious virus, as a proof-of-principle, affordable JEV vaccine.

## 2. Results

### 2.1. Development of Soluble CTB and LTB Fusion Proteins

JEV EDIII dimers are packaged along the five-fold symmetry into an icosahedral assembly on the virion surface. The fusion protein construct was designed to mimic the pentameric interface of JEV using CTB or LTB as a pentameric scaffold at the N-terminus of JEV EDIII (Figure 1A). The schematic modular view of the chimeric protein is shown in Figure 1B. We initially designed two different plasmids: CTB−JEV EDIII and LTB−JEV EDIII. CTB-JEV EDIII was amenable to soluble expression (Figure 1C). In contrast, LTB-JEV EDIII was resistant to soluble expression and was predominantly produced as highly insoluble inclusion bodies (Figure 1D). To overcome this problem, we constructed a third plasmid, hRID−LTB−JEV EDIII (Figure 1A), a tri-partite construct of a modular combination of hRID (RNA-interaction domain of human origin) as an RNA-mediated chaperone [27,28] at the N-terminal end of LTB−JEV EDIII. The TEV cleavage site was added to remove hRID if needed (Figure 1A). CTB and LTB fusion proteins were overexpressed in *E. coli* upon IPTG induction. CTB−JEV EDIII showed a solubility of 88.9% (Figure 1C,F). The LTB−JEV EDIII showed leaky expression in the absence of IPTG and resulted in predominantly insoluble pellet upon IPTG induction (Figure 1D). Of note, the solubility of LTB−JEV EDIII (2–3%) was dramatically increased up to 72% upon fusion with hRID (Figure 1D–F).

### 2.2. Assembly Status of CTB and LTB−JEV EDIII Chimeric Protein

To examine whether JEV EDIII self-assembled into a pentamer, we performed SEC and PAGE under non-reduced conditions. First, recombinant CTB and LTB fusion proteins were purified by nickel affinity chromatography. Purified CTB−EDIII and hRID−LTB-EDIII were 0.75 mg and 2.7 mg, respectively, from 500 mL *E. coli* culture medium (Appendix A). We then examined the assembly status of the purified protein using SEC. The first peak (void volume) shown in Figure 2A–C represents soluble aggregates. The second peak (estimated at 101.9 kDa) is comparable to the CTB−EDIII pentamer (expected size of 123 kDa). The size of hRID−LTB−EDIII was estimated at 150.7 kDa by SEC, comparable to the expected size of its pentameric assembly (168.3 kDa, Figure 2D). Each eluted fraction from SEC was analyzed by SDS−PAGE to validate the presence of fusion proteins (Figure 2A,B, right). hRID was cleaved from the hRID−LTB−EV by TEV protease (referred to as ‘LTB−JEV EDIII (TEV cut)’ hereafter). The estimated size by SEC analysis was 123.1 kDa, again comparable to the predicted pentameric size of 122.5 kDa. SDS−PAGE analysis with eluted fractions confirmed the monomeric size of LTB−EDIII (TEV cut) (Figure 2C, right). In addition to SEC analysis, we performed PAGE analysis under non-denaturing conditions. The monomeric band under denaturing conditions (boiling + DTT) shifted to the pentameric band (without boiling—DTT) (Figure 3A–C). These results clearly show that JEV EDIII fused with CTB or LTB self-assembles into a homopentamer.

### 2.3. Evaluation of Pentameric JEV EDIII

A biological evaluation of pentameric JEV−EDIII was conducted using GM1 binding assay. Both pentameric CTB and LTB bind to GM1, and their binding complex initiates endocytosis in the host cell [29,30]. Therefore, a test for the functional assembly of pentamers was performed by ELISA with ganglioside GM1 as the main receptor for binding. CTB−JEV EDIII and LTB−JEV EDIII (TEV cut) exhibited strong binding affinity toward GM1. Their binding capacity to GM1 was statistically comparable to the CTB−positive control under the same condition, whereas the LysRS negative control failed to bind (Figure 4A). Next, the stability of self-assembled pentamers was examined at various temperatures. CTB−EDIII and LTB−EDIII were incubated at different temperatures under reduced conditions (boiling, DTT). Then, the mobility shift of each pentamer was observed to determine the thermostability of EDIII fusion proteins. CTB−EDIII and LTB−EDIII (TEV cut) maintained their pentameric states below 65 °C. The intensity of the band corresponding to pentamers decreased when fusion proteins were exposed to 75 °C. Pentamers of both CTB− and LTB−EDIII failed to maintain their formation under harsher conditions at 100 °C (Figure 4B). Altogether, these results indicate the B subunit of CT and LT allows the formation of functional pentameric fusion proteins, validating their role as a pentameric scaffold for JEV EDIII.

### 2.4. Immunogenicity and Neutralization against JEV

To characterize the type of immune responses elicited by pentameric JEV E proteins, total levels of IgG, IgG1, and IgG2a subclasses were examined with sera from mice immunized three times. Balb/c mice were immunized with 25 μg of antigen three times, with a 2 week interval between vaccinations. Sera were isolated from mouse blood 2 weeks after each vaccination. Sera from mice immunized with CTB−EDIII or LTB−JEV EDIII (TEV cut) induced high levels of total IgG against the JEV Nakayama strain (Figure 5A). Both groups also showed high IgG1 levels similar to those of sera from mice immunized with an inactivated-JEV (i-JEV) vaccine (at half the human dose) (Figure 5B). Furthermore, sera from CTB−EDIII and LTB−JEV EDIII (TEV cut) immunized mice exhibited high levels of IgG2a. The i-JEV vaccine stimulated more IgG2a than the recombinant JEV EDIII fusion proteins (Figure 5B) but the difference was statistically insignificant. Despite having less of an impact on IgG2a class switching (Figure 5D), CTB− and LTB−EDIII vaccinations showed substantially high levels of neutralizing antibodies, similar to those of i-JEV. The plaque reduction neutralization tests (PRNT) were performed to assess neutralizing antibody (Nab) titers against the JEV Nakayama strain. Sera from mice immunized with CTB or LTB alone did not show Nab titers, but sera from mice vaccinated with CTB−EDIII or LTB−JEV EDIII (TEV cut) exhibited noticably high levels of Nab. The level of Nab induced by EDIII pentamers was comparable to that of Nab titers induced by the i-JEV vaccine (at a tenth of the human dose) (Figure 5C). 

## 3. Discussion

Unlike traditional vaccines produced by virus culture, live attenuated and inactivated recombinant protein vaccines are less immunostimulatory despite having higher safety profiles. Therefore, the ideal design of a recombinant vaccine would require strong adjuvants or carrier molecules. Alternatively, immunogenicity (and protective efficacy) could be enhanced by increasing the size of the antigen by assembling the monomers into a higher-order structure, preferably a virus-like particle (VLP) [31] or a nanoparticle (NP) [32,33,34]. In contrast to VLP/NPs, which primarily display major antigens, the present chimeric system displays both viral antigens and adjuvants at the same time, augmenting the protective immune response via a built-in-adjuvant function. Moreover, the *E. coli*-based expression system enables rapid production with low cost and ease of scale-up, making it a versatile and efficient system for producing recombinant antigen into self-assembled higher-order units. Unfortunately, designing such a vaccine, one that provides an adjuvant function while maintaining the necessary structural scaffold and thus preserving the conformational constraints for protective epitopes, has remained challenging [35,36]. 

Here, we successfully demonstrated the production of pentameric EDIII proteins using the non-toxic B5 component of the AB_5_ holo-toxin family as a stabilizing scaffold in an *E. coli* expression system. The formation of JEV EDIII pentamers was confirmed using biophysical tools, including SEC and gel analyses, under non-reduced conditions. The biological function of CTB/LTB fusion proteins was further validated by the GM1 receptor binding assay. Unlike CTB−EDIII, LTB−EDIII was resistant to soluble expression, which is commonly observed in bacterial expression [36]. The problem of misfolding into inclusion body formation was circumvented by a novel RNA-based chaperone system (chaperna) [37]. Extensive previous research identified the role of hRID as an RNA-mediated chaperone [27,28,37,38,39,40,41,42,43]. Most chimeric proteins combine different functional domains [28] or mutations in order to provide thermodynamic stabilization [44,45] but to the detriment of the kinetic folding process, resulting in extremely low yields during expression, if not insoluble aggregates. Pivotal to a successful recombinant chimeric antigen is, therefore, the harnessing of a chaperone-based design that would overcome the “kinetic trap” into a misfolded, immunologically irrelevant conformation. The hRID corresponding to the N-terminal appendage of human lysyl-tRNA synthetase, when placed at the N-terminal of LTB−EDIII, functions as an intra-molecular chaperone by interacting with tRNAs and restoring the solubility of monomers and their subsequent assembly into a functionally relevant pentamer structure [37]. There have been numerous attempts to produce pentameric CTB fusion proteins, either by eukaryotic expression or a bacterial system that requires additional steps of chemical refolding: hetero-pentameric CTB−JEV E fusion chimeric molecules from *P. pastoris* [46] or homopentameric CTB−JEV E fusion proteins in *E. coli* [47]. Our approach here extends and further improves the production process; coupled with an RNA-dependent chaperone system, the CTB−EDIII chimeric protein was soluble and pre-assembled as a stable, functional pentameric form without requiring ancillary steps of refolding from inclusion bodies. Beyond the biophysical methods adopted for evaluating the assembly status, e.g., non-denaturing SDS−PAGE and SEC, further analysis would be required for validating the assembly status of the EDIII domain at the atomic level, e.g., by cryo-EM or X-ray crystallography. Our design would serve as a universal pentameric blueprint for vaccine antigens across the flavivirus family of viruses, which include Zika, West Nile, and Dengue viruses (data not shown), as well as other viruses of icosahedral virion structure [48].

The potential immunostimulatory effects of recombinant EDIII pentamers were examined by serological assays, including total IgG, IgG class isotyping, and neutralizing antibody titers. Based on ELISA, the total IgG antibodies were greatly stimulated by the fusion with CTB/LTB, comparable to those of commercially available i-JEV-vaccinated mice. The difference in IgG levels between groups immunized with i-JEV and CTB/LTB fusion EDIII pentamers was not statistically significant. The stimulation may be due, in part, to the intrinsic adjuvant function of CTB/LTB [49] or, alternatively, to immunologically relevant epitopes on the pentameric configuration of the EDIII domain [50]. Moreover, the assembly of monomeric antigens into a highly repetitive structure effectively enlarges the antigen size, making it more recognizable by immune cells. Consistent with this view, epitopes presented on multivalent and highly repetitive outer structures led to cross-linking of B cell receptors, stimulating B cells, and inducing potent and long-lasting antibody responses [9,51]. The immune stimulatory effect is also reflected in the neutralization titer, as examined by in vitro PRNT_50_ assay. Nab titers triggered by CTB− and LTB−JEV EDIII were almost comparable to those achieved by the cell-culture-based i-JEV vaccine (Figure 5C). The lack of NT titer in the LTB/CTB control confirmed that the virus neutralization by the chimeric fusion protein was indeed due to the EDIII domain having assembled into an immunologically competent form. The present results merit challenge studies with JEV in susceptible animal models. The IgG2a/IgG1 ratio in immune sera was also examined as a signature for the Th1/Th2-type response [52,53,54], and no significant differences were observed between i-JEV and the CTB/LTB−EDIII recombinant antigens (Figure 5D). Parallel studies in our lab with CTB−Dengue EDIII pentamers showed that the IgG2a/IgG1 ratio was greatly increased by CTB fusion (data not shown), suggesting that the adjuvant effect of CTB cannot be generalized but rather is antigen-specific—either immunostimulatory or immunomodulatory, depending on the particular antigen combination [24]. 

Representing the non-toxic B-subunit of AB_5_-type holotoxins, CTB is a homopentameric protein responsible for binding to GM1 receptors, serving as a potent mucosal adjuvant with potential clinical applications [21]. The GM1 receptors are not exclusively located in epithelial cells; rather, they are present in all nucleated mammalian cells, including antigen-presenting cells [55,56,57]. This means that their immunostimulatory and adjuvant effects can be achieved not only by oral and nasal administrations but also via the intraperitoneal route. Further testing is needed to determine whether CTB/LTB pentameric EDIIIs formulated with different types of adjuvants and different vaccination routes trigger better immune responses than the alum-adjuvanted EDIII in the present study. 

In conclusion, our present study has shown that the recombinant viral subunit vaccine containing pentameric JEV EDIII is highly immunogenic, equivalent to a conventional i-JEV vaccine. Our pentameric design serves as a prototype for recombinant vaccines with antigens from across the Flavivirus genus and other icosahedral virons. The present system shows that AB_5_-type toxins as structural scaffolds with built-in adjuvants for recombinant vaccines could serve as a universal platform for sub-virion assembly.

## 4. Conclusions

The self-assembled pentameric JEV EDIII proteins were fabricated through one-step genetic conjugation in *E. coli*. To emulate the EDIII interface, which is a potent epitope on the virion surface, the B subunit of AB5 toxin was used as a pentamer structural scaffold. The CTB/LTB- JEV EDIII showed high immunogenicity, similar to that of conventional i-JEV in mice. Our proposed pentameric approach serves as a universal platform for recombinant vaccine antigens from across the Flavivirus genus and other icosahedral virions.

## 5. Materials and Methods

### 5.1. Vectors for Bacterial Recombinant Proteins

The pGE-LysRS vector [38] containing multi-cloning sites (KpnI, BamHI, EcoRV, SalI, HindIII) was used to generate CTB-JEV EDIII and LTB-JEV EDIII fusion proteins. CTB or LTB was inserted into the pGE-LysRS by NdeI/EcoRV after removal of LysRS. Then, EDIII was inserted into either pGE-CTB or pGE-LTB vectors using EcoRV and SalI. Then, 6x-histidine tags and a stop codon were added after the HindIII site. The pGE-hRID vector originated from pGE-LysRS. hRID-TEV cleavage sites and 6x-histidine tags were inserted into the pGE-LysRS vector treated with NdeI/BamHI. Then, LTB was cloned into the pGE-hRID using BamHI/EcoRV. EDIII and a stop codon were added to the vector treated with EcoRV/SalI. 

### 5.2. Protein Expression and Purification

The *E. coli* host Shuffle^®^ T7 (New England Biolabs, Inc., Ipswich, MA, USA), was transformed with a cloned expression vector. The cells were cultured in 500 mL of LB medium containing 50 μg/mL of ampicillin at 30 °C until the optical density (OD) at 600 nm was 0.5. Proteins were overexpressed by 1 mM isopropyl β-D-1-thiogalactopyranoside (IPTG) treatment at 20 °C. After 5 h of reactions, cells were harvested by centrifugation (3000× *g* for 20 min). Harvested cells were lysed via sonification in a lysis buffer of 50 mM Tris-HCl, 200 or 300 mM NaCl, and 10% glycerol at pH 7.0. The cell lysates were filtered through a polyether-sulfone syringe filter with a 0.45 μm pore size (Hyundai Micro Co., Ltd., Seoul, Republic of Korea) after centrifugation at 16,000× *g* for 10 min and purification via Ni-affinity chromatography (Bio-Scale™ Mini Nuvia™ IMAC Cartridges; Bio-Rad Laboratories, Inc., Hercules, CA, USA). The purified samples were concentrated with Centriprep™ (Merck Millipore Ltd., Tullagreen, Carrigtwohill, County Cork, Ireland). Protein samples were further separated by their size through size exclusion chromatography (SEC) with a Superdex™ 200 Increase 10/300 GL column (GE Healthcare, Uppsala, Sweden). After SEC, eluted fractions of the expected peak were pooled and concentrated with Centriprep™ (Merck Millipore Ltd., Tullagreen, Carrigtwohill, County Cork, Ireland). 

### 5.3. ELISA

The Nunc 96-well immunoplates (Thermo Fisher Scientific) were coated with antigens of interest at 4 °C and kept overnight. After washing with TBST, the plates were blocked with 1% BSA at room temperature for 2 h. Sera from immunized mice were serially diluted by 2-fold and applied to the coated plates for 1 h. After washing with TBST, an HRP-conjugated goat anti-mouse IgG antibody (Qiagen) was applied to the coated plate for 1 h at room temperature. Then, TMB (3,3′, 5,5′ tetramethylbenzidine) substrate reagent (BD Biosciences) was applied for 30 min. After 30 min, 2N H_2_SO_4_ was added, and the OD at 450 nm was measured by an ELISA reader (BMG LABTECH). 

### 5.4. GM1 Binding Assay

The Nunc 96-well microtiter immunoplates (Thermo Fisher Scientific) were coated with 300 ng/well of GM1 ganglioside (Sigma) diluted in a carbonate-bicarbonate buffer (pH 9.6; Sigma) and incubated at 4 °C overnight. The plates were washed with TBST with 0.05% Tween 20 and blocked with 1% bovine serum albumin (BSA) in PBST for 3 h at room temperature. The recombinant protein samples were 2-fold serially diluted from 300 ng with PBST, then applied to GM1-coated wells for 4 h at room temperature. Primary anti-penta-His antibody (100 μL/well; Qiagen) and secondary antibody conjugated with HRP (100 μL/well; Bethyl Laboratories) were added for 1 h at room temperature. The plates were washed 3 times with TBST and incubated with TMB (BD Biosciences) for 30 min, and the reaction was ended upon 2N H_2_SO_4_ addition. The optical density was measured at 450 nm via an ELISA reader.

### 5.5. Mouse Immunization and Serum Isolation

Groups of 5 female 6-week-old Balb/c mice were immunized intravenously with 25 μg of the recombinant proteins purified by SEC and their endotoxin removed (<10 EU/mL) by Triton X-114. Antigens were formulated with alum (Thermo Scientific) according to the manufacturer’s protocol. The mice were immunized 3 times, each with a 2-week interval. Blood was collected via ocular bleeding 2 weeks post-immunization and stabilized overnight at 4 °C prior to serum isolation by centrifugation (10,000 rpm, 30 min). 

### 5.6. Inactivated JEV Vaccine (i-JEV)

The commercially available i-JEV vaccine, Gcc-JEV Vaccine^®^, was purchased from Green Cross (GC). For ages under 3 years, 0.5 mL of the 1 mg/mL vaccine is recommended for a single dose. The recommended vaccination schedule is 3 times with a 1–2-week interval. Individuals above the age of 6 years should be immunized with 1 mL. The mice were immunized with the i-JEV vaccine at a 1/2- or 1/10-diluted dose for humans of 0.5 mg and 0.1 mg, respectively.

### 5.7. Cells and Viruses

JEV (Nakayama strain) was propagated in Vero cells. Vero cells were cultured at 37 °C in modified Eagle’s medium supplemented with 10% FBS. Cells were inoculated with 0.01 multiplicity of infection of JEV for 1.5 h. JEV was harvested 2–3 days post-inoculation. The harvested viruses were concentrated using an Amicon^®^ Ultra-15 Centrifugal Filter Unit (Merk) when necessary. A plaque assay was performed for viral titration. JEV was kindly provided by the International Vaccine Institute.

### 5.8. Plaque Assay

Vero cells were seeded on 6-well plates. Monolayered Vero cells were washed with PBS before viral absorption. Each well was inoculated with a 10-fold serially diluted virus for 1.5 h. After inoculation, cells were washed with PBS and a 1% agarose overlay was applied. The cells were incubated for 2–3 days and subjected to plaque counting.

### 5.9. Neutralization Assay

A plaque-reduction neutralization assay was performed with Vero cells in 6-well plates. Sera from immunized mice were inactivated by 56 °C heat inactivation for 30 min. Sera were 2-fold serially diluted in serum-free MEM. The JEV was prepared for 100 pfu/well in serum-free MEM. The virus was mixed in equal volumes with inactivated sera. The mixtures were incubated at 37 °C in 5% CO_2_ for 1 h, applied to Vero cells in 6-well plates, and absorbed for 90 min. After virus adsorption, cells were washed with PBS and coated with a 1% agarose overlay. The plaques were counted after 72 h of incubation. The plaque-reduction neutralization titer, PRNT_50_, was calculated as the highest dilution titer, with a 50% reduction in the number of plaques compared to serum-free controls. 

### 5.10. Statistical Analysis

All data are presented as means, scatter plots, or error bars. The error bar indicates the standard deviation. The Student’s *t*-test was conducted to compare two different groups, and a one-way ANOVA using the Tukey multiple comparison test was conducted to analyze 3 or more groups. The value was considered statistically significant when *p* < 0.05 (*** indicates *p* < 0.001, ** indicates *p* < 0.01, * indicates *p* < 0.05, n.s. indicates statistically non-significant).

## Figures and Tables

**Figure 1 toxins-15-00425-f001:**
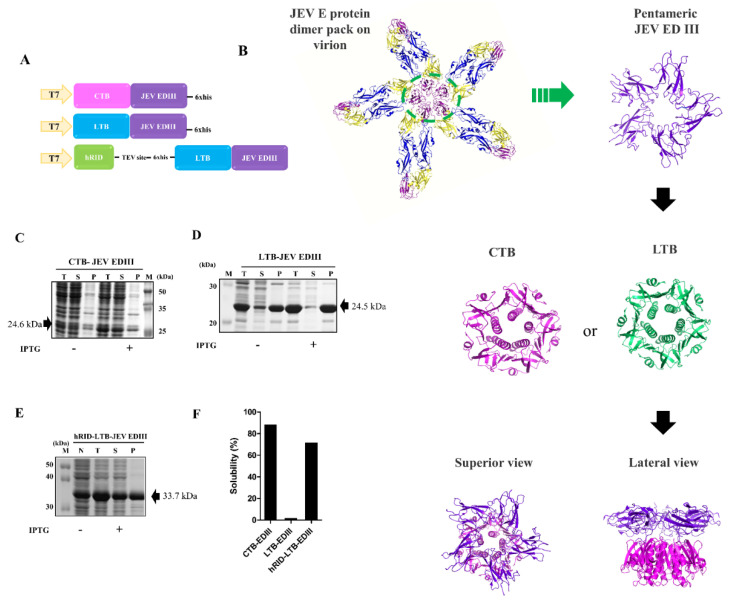
Development of recombinant CTB−/LTB−EDIII fusion proteins. (**A**) Three versions of the JEV EDIII gene construct. (**B**) Schematic diagrams of the JEV EDIII pentamer. (CTB: 2CHB pdb, LTB: 1FD7 pdb, JEV EDIII 5WSN pdb). (**C**–**E**) Protein expression levels and solubility of CTB−JEV EDIII (**C**), LTB−JEV EDIII (**D**), and hRID−LTB−JEV EDIII (**E**). The expected molecular size of CTB−JEV EDIII, LTB−JEV EDIII, and hRID−LTB−JEV EDIII are 24.6 kDa, 24.5 kDa, and 33.7 kDa, respectively. (**F**) Solubility percentiles of the three recombinant proteins.

**Figure 2 toxins-15-00425-f002:**
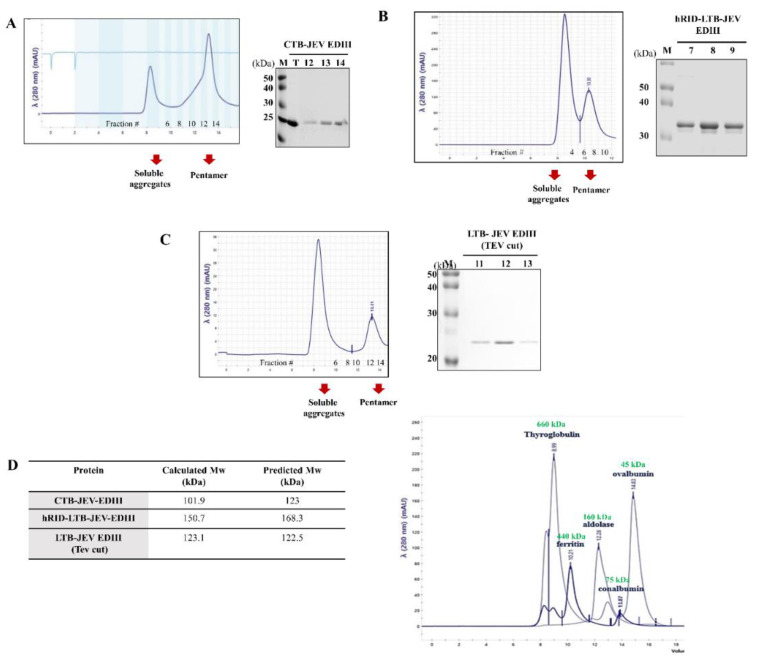
Evaluation of pentamer formation via SEC. SDS−PAGE analysis of eluted fractions of CTB−JEV EDIII (**A**), hRID−LTB−JEV EDIII (**B**), and LTB−JEV EDIII (**C**). (**D**) Table (left) representing the measured (middle) and estimated (right) sizes of the three EDIII pentamers (left). Molecular weight was calculated by SEC with the calibration proteins in the absorbance plot (right).

**Figure 3 toxins-15-00425-f003:**
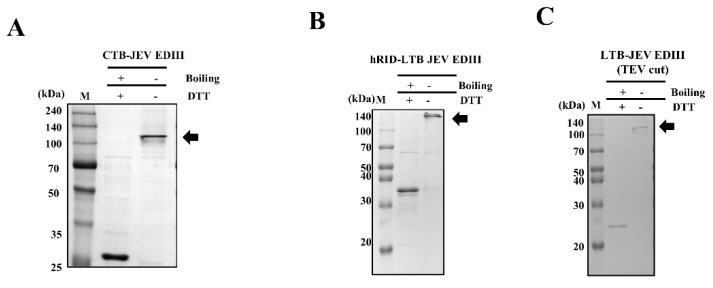
Validation of pentamer formation via gel analysis. Representative gel images of **CTB**−**EDIII** (**A**), hRID−LTB JEV EDIII (**B**), and LTB−JEV EDIII (TEV cut) (**C**), demonstrating a shift in protein band under non-reduced conditions. The molecular size of CTB and hRID−LTB JEV EDIII monomer and pentamer were 24.6/33.7 kDa and 123/168.3 kDa, respectively. The molecular size of LTB−JEV EDIII (TEV cut) monomer pentamer were 24.5 kDa and 122.5 kDa, respectively. Each protein was loaded onto the gel, followed by SEC. The symbols + and − represent with or without the DTT and boiling step. Black arrows point to the expected size of each pentamer.

**Figure 4 toxins-15-00425-f004:**
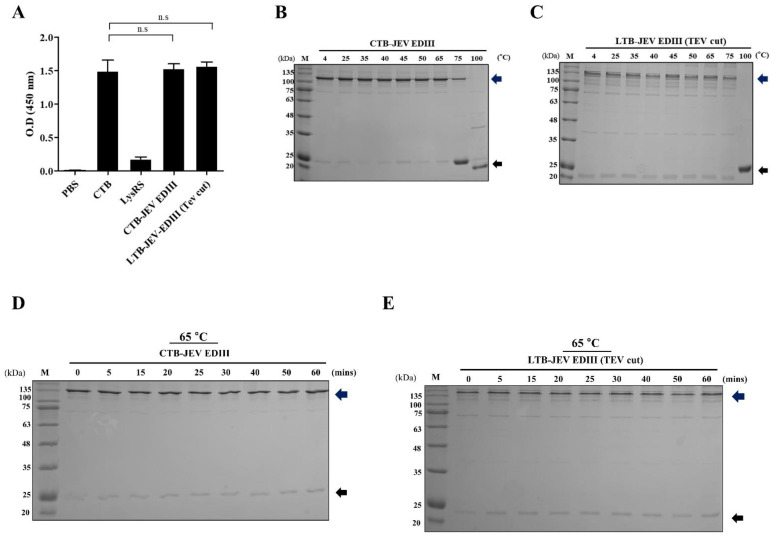
Evaluation of JEV EDIII pentamers. (**A**) GM1 ganglioside binding assay to determine pentamers JEV EDIII. GM1 was coating antigen and binding affinity against recombinant fusion proteins. pLys represents LysRS protein as negative control. The comparison of each column with PBS control was analyzed by a one-way ANOVA with a post-hoc Tukey’s multiple comparison test. (n.s: not significant) (**B**,**C**) Stability test of pentameric CTB−/LTB−JEV EDIII. Fusion proteins were incubated for 15 min at different temperatures indicated at the top of each gel column. (**D**,**E**) Thermostability test of pentameric JEV EDII fusion proteins. Fusion proteins were incubated at the fixed temperature of 65 °C at various time points indicated at the top of each gel column.

**Figure 5 toxins-15-00425-f005:**
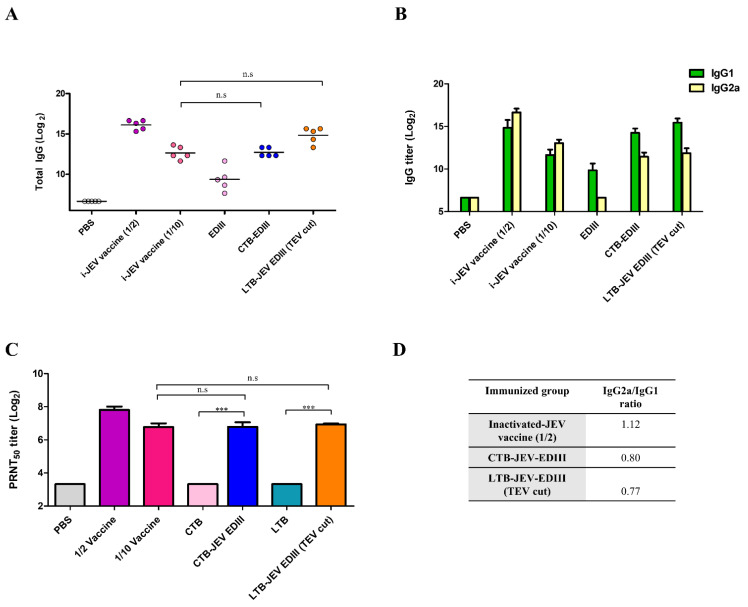
Immunogenicity of CTB−JEV EDIII proteins against the JEV Nakayama strain. JEV-specific total IgG (**A**), IgG1 and IgG2a (**B**) levels in immunized Balb/c mice obtained via ELISA (n = 5). (**C**) PRNT_50_ a ssay against JEV Nakayama. (**D**) Table summary of IgG2a/IgG1 ratio obtained from the three immunization groups. Student’s *t*-test used to compare two different groups. One-way ANOVA followed by Turkey’s multiple comparison test was conducted to compare three or more groups. (*** *p* < 0.001; n.s: not significant).

## Data Availability

The data presented in this study are available in this article or Appendix A.

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
