# Peer review of "AB5-Type Toxin as a Pentameric Scaffold in Recombinant Vaccines against the Japanese Encephalitis Virus"

_toxins, 2023, doi:10.3390/toxins15070425_

Round 1

Reviewer 1 Report

The paper tests the idea that assembly of the recombinant JE E protein domain EDIII into its native pentameric conformation would provide enhanced antigenicity to this viral target epitope – especially when coupled covalently to a vaccine adjuvant. To do this, the authors use the pentameric assembly of CTx and LTx B-subunits – by fusing the EDIII domain at the C-termiuns of each CTX or LTx B-subunit monomer. They successfully express the fusion proteins and show expression as pentamers. They then show function of the CTx and LTx B-subunit domains as evidenced by binding GM1 in vitro. When injected IV for immunization the fusion molecule pentamers induce an enhanced Ig-antibody response that can block viral invasion when tested in vitro (plaque assay).

The studies are well conceived and technically well done – but the main idea, the novel claim, that pentameric assembly of the EDIII domain (i.e. “sub-viral assembly”) would lead to enhanced immunogenicity, has not been conclusively tested. The amplified immunogenicity seen in the fusion pentamers could be explained entirely by its fusion to the known vaccine adjuvants and delivery vehicles CTxB and LTxB.  Suggest that the authors test the pentameric assembly idea by creating hetero-pentamers of CTxB and CTxB-EDIII fusion monomers (thus breaking apart the EDIII pentamer domain) and test those molecules for immunogenicity. This can be done by expressing the native and fusion B-subunits in E. coli – urea denature the pentamers, and then allow reassembly under different stoichiometries of native vs fusion B-monomers by dialysis into physiologic buffer.

Many of the figure legends need more complete explanations – column and some panel legends missing.

Fig 5: This reader does not see the 60-fold enhancement of Ig-response caused by the fusion molecules as claimed (discussion section)

Author Response

We highly appreciate the reviewer's comment. 

Reviewer 2 Report

The draft is quite informative and import piece of work. Before the draft is officially accepted for publication, author need to address following coments

Author should defined the LMICs before directly using the abbreviation

In suppl. figure legend, author mentioned 8-15 % agarose gel. Is it correct?? it should polyacrylamide gel electrophoresis (SDS-PAGE)

Author should also explain why this platform is better than VLPs and should include or cite recent paper discussing benefits of VLPs (see https://www.mdpi.com/2076-393X/11/2/479). Author should also mention why bacterial system is better as this system have problem of endotoxin

In line 360, italic the E. coli

In line 316, s6-week????

In fig 1C, author should mark or point protein band induced or expected

Author should mention the mass of each fusion protein, difficult to really understand data in figure

Surprisingly, pattern of bands in protein marker is quite different in gel image in fig1c and fig1d, are these two different protein markers?????   In fig 1d, protein induction even in absence of IPTG????, quite surprising. Same in figure 1e
Standard error bar or deviation missing from figure 1f Data in figure 1e and 1f not matching  

Draft need minor check on writing issues  

Draft require minor writing attention

Author Response

(The authors gave the same response as above.)

Round 2

Reviewer 1 Report

the paper is improved

Reviewer 2 Report

Since author responded to the comments in sufficient detail, the manuscript can be accepted in its present from.

Over all writing is OK, just need minor editing